Corrected: Publisher correction

# A network of chaperones prevents and detects failures in membrane protein lipid bilayer integration

João P.L. Coelho[1], Matthias Stahl[1,4], Nicolas Bloemeke[1], Kevin Meighen-Berger[1], Carlos Piedrafita Alvira[1], Zai-Rong Zhang[2], Stephan A. Sieber [1] & Matthias J. Feige[1,3]

A fundamental step in membrane protein biogenesis is their integration into the lipid bilayer with a defined orientation of each transmembrane segment. Despite this, it remains unclear how cells detect and handle failures in this process. Here we show that single point mutations in the membrane protein connexin 32 (Cx32), which cause Charcot-Marie-Tooth disease, can cause failures in membrane integration. This leads to Cx32 transport defects and rapid degradation. Our data show that multiple chaperones detect and remedy this aberrant behavior: the ER–membrane complex (EMC) aids in membrane integration of low-hydrophobicity transmembrane segments. If they fail to integrate, these are recognized by the ER–lumenal chaperone BiP. Ultimately, the E3 ligase gp78 ubiquitinates Cx32 proteins, targeting them for degradation. Thus, cells use a coordinated system of chaperones for the complex task of membrane protein biogenesis, which can be compromised by single point mutations, causing human disease.

[1] Center for Integrated Protein Science at the Department of Chemistry, Technical University of Munich, Lichtenbergstr. 4, 85748 Garching, Germany. [2] Interdisciplinary Research Center on Biology and Chemistry, Shanghai Institute of Organic Chemistry, Chinese Academy of Sciences, Shanghai 201210, China. [3] Institute for Advanced Study, Technical University of Munich, Lichtenbergstr. 2a, 85748 Garching, Germany. [4] Present address: SciLifeLab, Department of Oncology-Pathology, Karolinska Institutet, Box 1031171 21 Solna Stockholm, Sweden. Correspondence and requests for materials should be addressed to M.J.F. (email: matthias.feige@tum.de)

In complex organisms, a major part of the proteome consists of integral membrane proteins. Membrane proteins mediate transport processes and enable cellular communication and motility, ultimately allowing multicellular structures to exist. Like other eukaryotic proteins of the secretory pathway, membrane proteins are produced at the endoplasmic reticulum (ER). Their very nature, however, integration of hydrophobic transmembrane (TM) regions into the lipid bilayer, correct assembly of TM segments as well as folding of soluble domains, poses complex challenges to the ER quality control system. How the ER quality control system addresses these challenges and reliably scrutinizes the different steps of membrane protein biogenesis remains incompletely understood, as most work has focused on folding, quality control and degradation of soluble proteins in the ER.

A critical step in the biosynthesis of membrane proteins is their correct topogenesis, i.e., the integration of each TM helix in the adequate orientation into the lipid bilayer[1]. Whereas for simple, strongly hydrophobic TM regions, acting as stop-transfer sequences, the process of co-translational topogenesis is well-understood[2,3], these TM regions represent only a subset of all naturally occurring TM sequences. In fact, functional requirements very often compete with the structural stability of proteins[4], and membrane proteins are no exception to this rule: the hydrophobic interior of the lipid bilayer would energetically favor integration of nonpolar residues, but defined intra or intermolecular interactions of TM helices as well as functional requirements, e.g., providing a hydrophilic cavity in ion channels, can compromise membrane protein integration[5]. Indeed, it has been shown that for several multipass TM proteins, individual TM segments can be transiently exposed to either the ER lumen or cytoplasm, and that correct topogenesis only occurs post-translationally, mediated by TM segment interactions[6–11]. A similar behavior has been described for single-pass TM proteins of low hydrophobicity, where intermolecular interactions may be coupled to proper membrane integration[12–14]. Accordingly, membrane protein topogenesis and TM helix integration are intimately linked to the correct assembly of individual TM helices within the membrane[15,16].

Individual TM helices that are energetically unfavorable in the membrane are not a rare phenomenon: ca. 25% of all TM helices in multipass proteins, when analyzed in isolation, are predicted to have an unfavorable free energy for membrane integration[17]. For these, dependence on defined TM–TM interactions for correct integration and topogenesis can be expected. If correct membrane integration and topogenesis depend on the interaction of TM regions dispersed throughout a protein, it immediately follows that this process can fail, like any protein folding and assembly reaction. It is thus likely that the cell has developed means to identify and either correct or dispose of incorrectly integrated TM proteins. These mechanisms remain mostly unknown, but they are particularly relevant since polar residues due to mutations in TM regions are very often associated with human disease[18,19]. An important example is connexin 32 (Cx32), which is the focus of this study. Cx32 is a four-helix TM protein that forms homo-hexamers, connexons, within the plasma membrane. Subsequently, two hexameric connexons embedded within different membranes dock onto each other to form a channel, many of which constitute gap junctions[20,21]. Cx32-formed gap junctions are essential for transport of small molecules through the multiple isolating membrane layers Schwann cells build around nerve cell axons. Mutations in Cx32 can thus lead to demyelination and cause X-linked Charcot–Marie–Tooth disease, a prevalent disorder of the peripheral nervous system[22].

Using Cx32 as a biomedically relevant model system, within this study we addressed the question if and how the ER quality control system can prevent, recognize and handle aberrant membrane protein lipid bilayer integration and topogenesis.

## Results

**Point mutations in Cx32 can cause membrane misintegration.** The gap-junction forming protein Cx32 contains four TM helices, with its N- and C-termini being exposed to the cytoplasm (Fig. 1a). Within this work, we were interested in the question if and how aberrant topogenesis may occur in mutants of Cx32. We, therefore, selected four disease-causing mutations within Cx32, one in each of its TM helices, that were all predicted to significantly destabilize integration of the respective helix (Fig. 1b). Next, to define the exact location of each mutation, we generated a homology model of Cx32, based on the crystal structure of Cx26[21]. When analyzed within the modeled structure of the hexameric Cx32 connexon, all of the four mutations were predicted to be located within the membrane (Fig. 1c). For two of the mutants, A147D and I203N, an exposure to the outside of the connexon, i.e., the lipid environment, was predicted. For the other two mutants, M34K and L90H, a location within the Cx32 interfaces that make up the connexon was observed in the model (Fig. 1c).

To assess the structural effects of the different mutations, we developed a glycosylation reporter system inspired by previous work[23]. We individually included an artificial glycosylation site (NVT) into each of the three Cx32 loops connecting its four TM helices as well as its C-terminal tail, flanked by $(GS)_2$ sequences (Supplementary Fig. 1a). No artificial glycosylation site was included into the N-terminal tail of Cx32, which explains its slightly faster migration on sodium dodecyl sulfate polyacrylamide gel electrophoresis (SDS-PAGE) gels (Fig. 1d), as it naturally contains an NWT glycosylation site directly after the initiation methionine (Supplementary Fig. 1a). Constructs were made for wild-type Cx32 as well as its disease-causing mutants M34K, L90H, A147D, and I203N (Fig. 1d, e and Supplementary Fig. 1b–d). Since the ER is the only organelle where glycosylation of the aforementioned sites can occur, this assay allows to deduce the topology of the protein under investigation. For wild type Cx32 (Cx32wt) we observed glycosylation of sites within loops 1 and 3, and no glycosylation of either the N-terminal or C-terminal site or a site within loop 2 (Fig. 1d). These findings are in agreement with the predicted topology of Cx32wt (Fig. 1a). Next, we used the same assay to assess the topology of the disease-causing mutants Cx32M34K, Cx32L90H, Cx32A147D, and Cx32I203N. For Cx32M34K, Cx32A147D and Cx32I203N, we observed the same glycosylation pattern as for Cx32wt, arguing that the topology of these mutants was unaltered (Supplementary Fig. 1b–d). In contrast, for Cx32L90H we detected partial glycosylation of the reporter site in loop 2 (Fig. 1e), arguing for its (partial) exposure to the ER lumen. All other reporter sites were unaltered in their glycosylation behavior for this mutant. These findings suggest that Cx32L90H can adopt an altered topology, with TM helices 2 and 3 becoming exposed to the ER lumen (Fig. 1e). Of note, Cx32L90H has the most unfavorable predicted free energy of membrane integration among all four mutants tested (Fig. 1b). To assess if our findings were more general or limited to Cx32L90H, we analyzed two additional disease-associated mutants of Cx32, both affecting TM helix 2: Cx32L81H and Cx32L83R. In the modeled structure of Cx32, residue L83 was part of the connexon interface like L90, whereas L81 was predicted to be located in the core of a single Cx32 monomer (Supplementary Fig. 1e). Both of these mutants also showed glycosylation of a loop 2 reporter site, suggesting that mutations in the second TM helix of Cx32 can generally lead to membrane misintegration and failures in topogenesis (Fig. 1f).

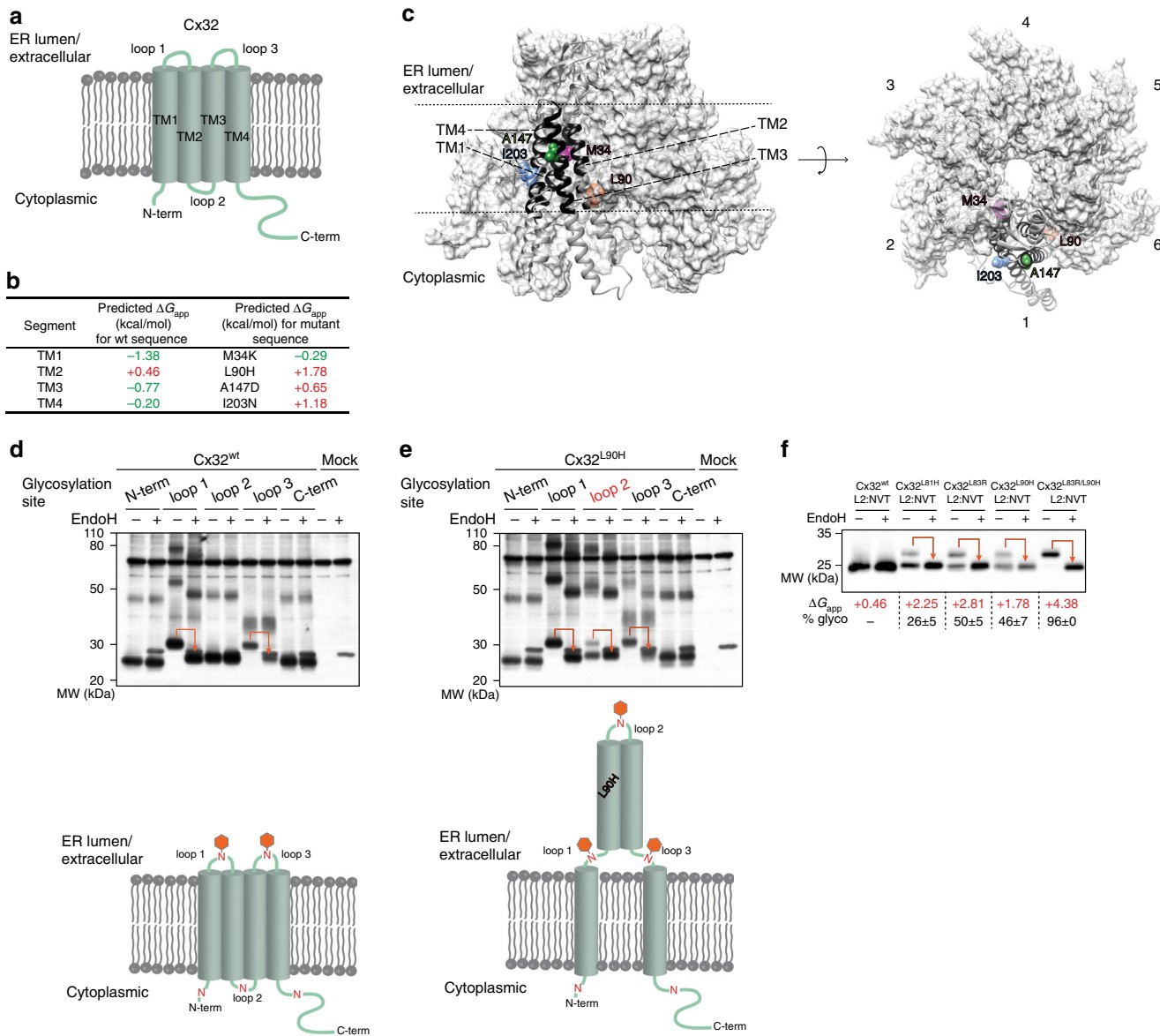

**Fig. 1** Single-point mutations lead to failures in membrane integration for Cx32. **a** Schematic of Cx32, showing its predicted topology. **b** Predicted free energies for helix insertion for apolar-to-polar missense mutations investigated in this study. For $\Delta G < 0$ (green) an energetically favorable membrane integration is predicted, while $\Delta G > 0$ (red) indicates an unfavorable insertion reaction. Transmembrane helices and $\Delta G$ values were predicted according to Hessa et al.[17]. **c** Side and top view of the modeled hexameric Cx32 connexon. Disease-causing mutants investigated in this study are shown in a CPK representation on a single monomer. Transmembrane helices are shown in black. Individual Cx32 monomers are numbered from 1 to 6. **d** Cx32[wt] and **e** Cx32[L90H] with individually introduced glycosylation sites in the indicated regions were transfected into HEK293T cells, lysates treated with or without EndoH as indicated and analyzed by immunoblotting. The schematics below each indicate the location of the individually assessed glycosylation sites (N) and possible topologies deduced from the observed glycosylation. An orange arrow in the blots and an orange hexagon in the schematic indicate sites that became glycosylated. **f** HEK293T cells transfected with the indicated Cx32 TM2 mutant constructs were analyzed as in **d**, **e**. Constructs all carried a glycosylation reporter site in loop 2 (L2:NVT). Predicted free energies of membrane insertion for TM2 of the respective proteins as well as a quantification of relative loop 2 reporter site glycosylation (mean ± SEM, $N = 3$) are shown below the immunoblots

Despite their partially altered topologies, all mutants still formed disulfide bonds, which connect the extracellular loops in the wild type protein (Supplementary Fig. 1f), arguing against global misfolding.

Since we only observed partial glycosylation for the three Cx32 mutants Cx32[L81H], Cx32[L83R] and Cx32[L90H], we wondered if either incomplete exposure of their TM helices or simply incomplete glycosylation of the reporter sites was the explanation for this behavior. We thus combined two disease-causing mutations in TM helix 2 of Cx32 (L83R and L90H), which together resulted in a highly unfavorable predicted free energy

for membrane integration (Fig. 1f). Cx32[L83R, L90H] became almost quantitatively glycosylated at a reporter site within loop 2 (Fig. 1f), arguing that our assay properly reports on the degree of TM helix integration and that the single point mutations lead to partially altered topologies.

**Cx32 mutants are compromised in transport and stability.** Since we observed failures in membrane integration for Cx32[L90H], we wondered if the mutant would be recognized by ER quality control. We thus first assessed its subcellular localization by fluorescence microscopy in COS-7 cells. To this end, a

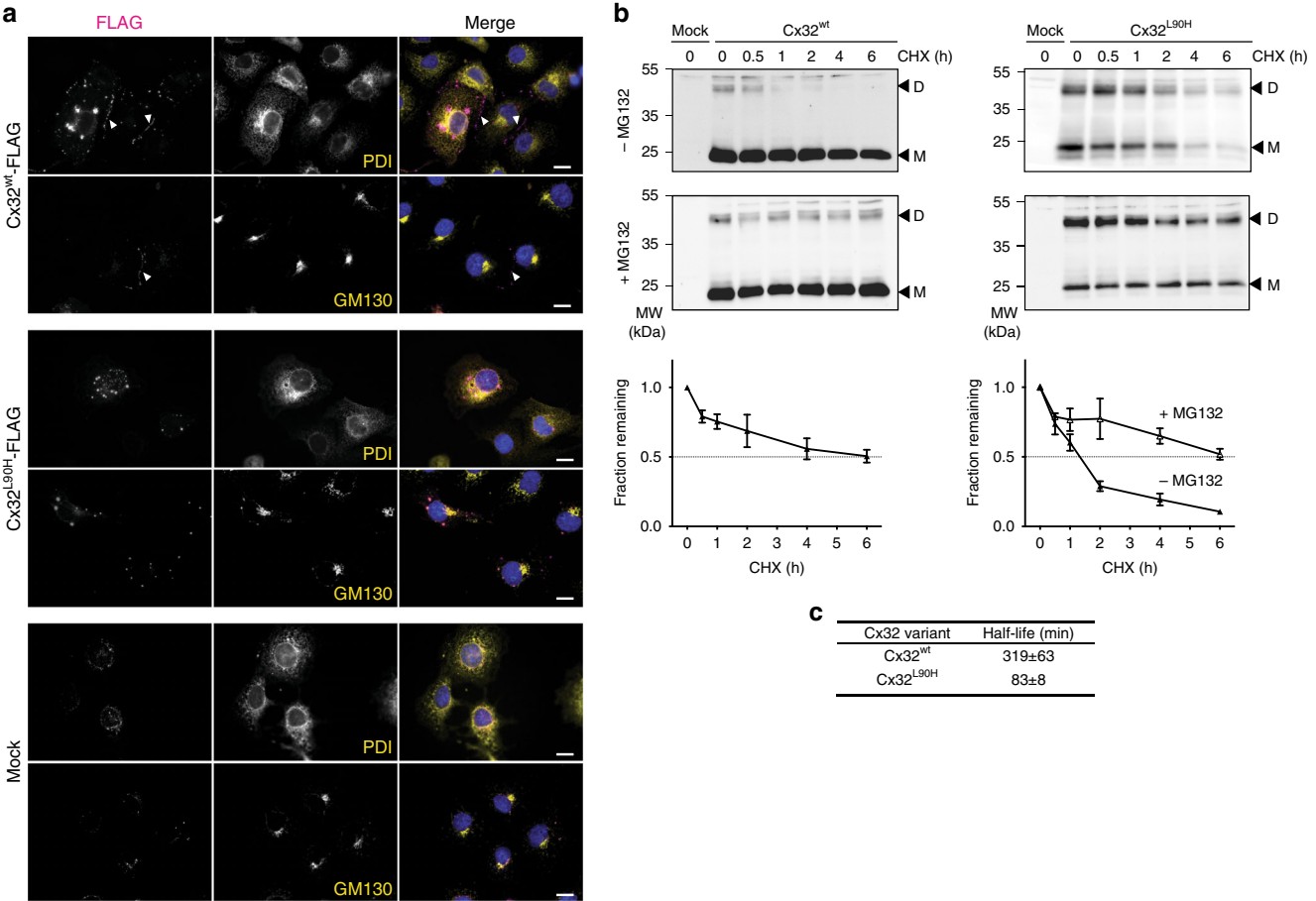

**Fig. 2** Cx32[L90H] shows defects in gap-junction formation and rapid degradation. **a** COS-7 cells were transfected with the indicated constructs and immunostained for FLAG-tagged Cx32 (magenta), PDI (yellow) as an ER marker, or GM130 (yellow) as a Golgi marker. Nuclei were stained with DAPI (blue). Anti-FLAG immunofluorescence data are depicted as maximum intensity projections from deconvoluted z-stacks, while PDI, GM130, and nuclei are shown as a central cell plane from the same, nondeconvoluted images. Gap-junction plates lining cell–cell boundaries are indicated with white arrowheads. Pictures are representative of cells from at least three different biological replicates. Scale bars correspond to 20 μm. **b** HEK293T cells transfected with the indicated constructs were incubated with either cycloheximide (CHX) alone, or additionally with the proteasome inhibitor MG132 where indicated. Arrowheads indicate monomer (M) and dimer (D) bands quantified to determine Cx32 turnover. Quantifications are shown below the immunoblots (mean ± SEM, $N \geq 3$). **c** Half-lifes without MG132 for Cx32[wt] and Cx32[L90H] as derived from **b** are shown

C-terminal FLAG tag was engineered to the proteins under investigation (Supplementary Fig. 1a). Microscopy experiments for Cx32[wt] revealed fluorescent punctae in transfected cells, frequently lining cell–cell boundaries (Fig. 2a). This behavior is typical of gap-junction plaques observed for connexin family members[24,25] and argues for proper transport and gap-junction formation of FLAG-tagged Cx32[wt]. Differences were observed for Cx32[L90H]: although Cx32[L90H] still formed some punctae, these were reduced in number in comparison to Cx32[wt] and appeared not to be localized at the typical cell–cell boundary observed for Cx32[wt] (Fig. 2a and Supplementary Fig. 2a). In agreement with our microscopy data, Cx32[L90H] did not form detergent-resistant species, which were observed for Cx32[wt] and are characteristic for gap-junction plaques[26] (Supplementary Fig. 2b). Despite this, we still observed EndoH-resistant, PNGaseF sensitive species for Cx32[L90H] glycosylated in loop 3 (Supplementary Fig. 2c). EndoH removes high mannose sugars that have not been further modified in the Golgi, whereas PNGaseF also removes Golgi-modified complex oligosaccharides. EndoH resistance would occur upon transport through the Golgi, which accordingly Cx32[L90H] is still able to reach. Since no co-localization of a Golgi marker and Cx32[L90H] was observed (Fig. 2a), Cx32[L90H] can most likely still traverse this organelle.

Taken together, these data argue that Cx32[L90H] can still pass the Golgi but is compromised in gap-junction formation. Species that are still transported are likely to be membrane integrated, which we found to occur for ca. 50% of Cx32[L90H] (Fig. 1f). To test this further, we also performed microscopy experiments on the Cx32[L81H] mutant, which showed transport but also partial ER retention, similar to Cx32[L90H]. In contrast, Cx32[L83R/L90H] was completely retained in the ER, corroborating our hypothesis that membrane misintegration correlates with ER retention (Supplementary Fig. 2d).

We next investigated if Cx32[L90H] would be recognized by quality control processes and, e.g., become a substrate of ER-associated degradation (ERAD). Indeed, whereas Cx32[wt] was degraded with a half-life of 5–6 h, Cx32[L90H] was degraded much faster with a half-life of only ca. 1.5 h (Fig. 2b, c and Supplementary Fig. 2e). The proteasome inhibitor MG132 blocked degradation of Cx32[L90H], corroborating its fate as an ERAD substrate (Fig. 2b, c and Supplementary Fig. 2e).

**Chaperone cooperation in membrane protein biogenesis.** Since we observed accelerated ERAD for Cx32[L90H] in comparison to Cx32[wt], Cx32[L90H] was apparently recognized as faulty by the ER quality control system. We thus aimed to determine whether

chaperone interactions in the ER were affected by the mutation, either qualitatively or quantitatively. To this end, we transfected FLAG-tagged Cx32$^{wt}$ or Cx32$^{L90H}$ into HEK293T cells, lysed cells in mild digitonin buffer, immunoprecipitated the Cx32 proteins and performed affinity-enrichment mass spectrometry experiments[27]. Whereas no ER chaperone interaction was either exclusively found for the wild type or the mutant, the mass spectrometry experiments revealed two interesting potential interaction partners of Cx32: the ER chaperone Calnexin (Cnx) and the ER–membrane protein complex (EMC) subunit 10 (EMC10) (Fig. 3a and Supplementary Fig. 3a, b). Cx32 is not a glycoprotein (Fig. 1d), arguing for glycan-independent

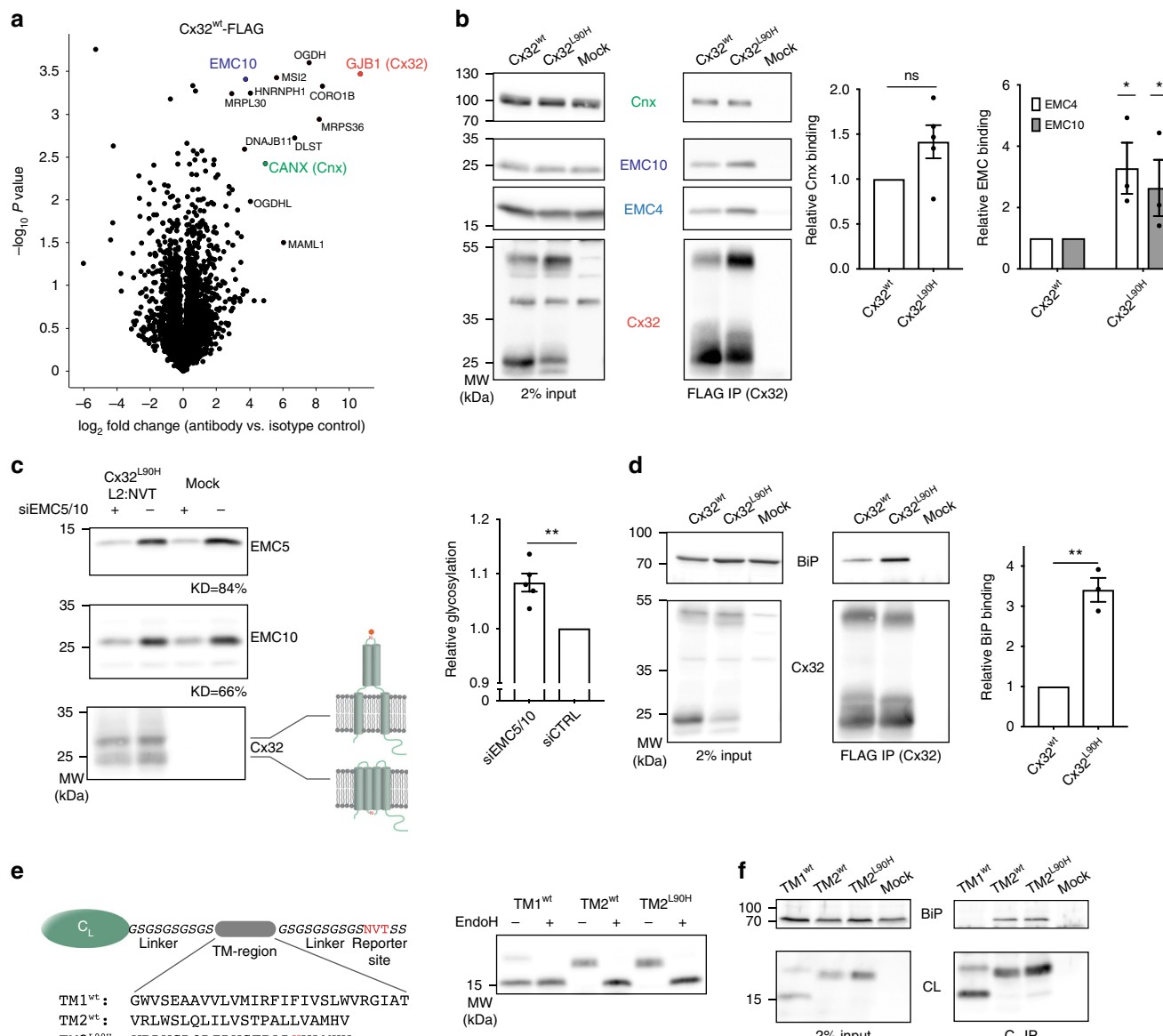

**Fig. 3** Connexin mutants are recognized by the ER quality control system. **a** Mass spectrometry volcano plot for FLAG-tagged Cx32$^{wt}$, immunoprecipitated in 1% digitonin from transfected HEK293T cells. Enriched proteins are denoted with their Uniprot gene name. Cx32 is shown in orange and ER chaperones investigated further in this study are highlighted in blue (EMC10) and green (Cnx). Either a rabbit monoclonal anti-FLAG antibody or a rabbit IgG isotype control was used. **b** Representative blots from immunoprecipitation experiments from HEK293T cells transfected with the indicated Cx32 constructs. Interaction of Cx32 with endogenous Cnx and EMC subunits was detected and increased for Cx32$^{L90H}$ with both EMC4 and EMC10 (mean ± SEM, $N \geq 3$, ns: nonsignificant, *$P$ value < 0.05, two-tailed Student's $t$ tests). Quantifications were performed as described in the Methods section. **c** Transient knockdown of EMC5/10 by siRNA (average knockdown (KD) efficiencies are shown below the blots) increases glycosylation of a reporter site in loop 2 for Cx32$^{L90H}$. Monomeric species ±glycosylation are shown on the blot, indicative of the topologies depicted on the right. Changes in glycosylation, quantified as described in the Methods section, are shown on the right (mean ± SEM, $N = 5$, **$P$ value < 0.01, two-tailed Student's $t$ tests). **d** Same as in **b** for co-transfected hamster BiP (mean ± SEM, $N = 3$, **$P$ value < 0.01, two-tailed Student's $t$ tests). **e** Schematic of a reporter construct to assess BiP-TM region binding. The nonglycosylated immunoglobulin λ light chain C$_L$ domain with its own ER import sequence is followed by a flexible linker connected to the TM sequence of interest shown below the schematic. TM segment 1 was inverted to allow for a type I topology. A C-terminal glycosylation site (NVT, marked in red) allows to assess membrane integration (no glycosylation) versus ER import (possible glycosylation). Membrane integration/ER import was assessed for Cx32 TM segment 1, 2, and 2 carrying the L90H mutation by transfection of the constructs into HEK293T cells and EndoH deglycosylation where indicated. **f** C$_L$-TM constructs were co-transfected with hamster BiP into HEK293T cells and their interaction was analyzed by co-immunoprecipitation experiments coupled to immunoblots

recognition of Cx32 by the lectin chaperone Cnx. Based on these data we assessed interaction of Cx32$^{wt}$ and Cx32$^{L90H}$ with endogenous Cnx. Co-immunoprecipitation experiments using FLAG-tagged Cx32 constructs revealed interaction for Cx32$^{wt}$ and Cx32$^{L90H}$ with only slightly stronger interaction of the mutant with Cnx (Fig. 3b). Next, we assessed Cx32 interaction with endogenous EMC. The EMC consists of 10 subunits (EMC1–10) in mammalian cells, seven of which are predicted integral membrane proteins (EMC1, 3–7, and 10) and three are cytoplasmic (EMC2, 8, and 9)[28,29]. The EMC is involved in ERAD[29,30], and has recently been shown to facilitate membrane integration of tail-anchored proteins with low-hydrophobicity TM regions[31]. It furthermore has been shown to be involved in the biogenesis of membrane proteins with unstable TM regions[32]. We thus wondered if its interaction with Cx32 was affected by disease-causing polar mutations in Cx32 TM regions which we had found to compromise membrane integration. Indeed, EMC4 and EMC10 showed significantly stronger interaction with Cx32$^{L90H}$ compared to Cx32$^{wt}$ (Fig. 3b). Increased EMC interaction was also observed for another mutant we had found to show membrane misintegration, Cx32$^{L81H}$ (Fig. 1f and Supplementary Fig. 4a). Of note, whereas Cx32$^{L90H}$ showed an increase in SDS-resistant dimers in comparison to Cx32$^{wt}$ (Fig. 3b), Cx32$^{L81H}$ was mostly monomeric on SDS-PAGE gels (Supplementary Fig. 4a,c), arguing against a dominant effect of client oligomerization on EMC interaction with Cx32.

Despite recent evidence for a participation of EMC in the biogenesis of tail-anchored and multipass TM proteins[31–34], direct biochemical effects on the latter are still unclear. We thus established a transient siRNA knockdown of EMC5 and 10, destabilizing the whole EMC[31], and assessed the effect on membrane integration of Cx32$^{wt}$ and the Cx32$^{L90H}$ mutant. Knockdown of EMC5/10 by ca. 65–85% led to a small but significant increase in glycosylation of the Cx32$^{L90H}$ mutant at the reporter site in loop 2 (Fig. 3c). Thus, although our observed effects were modest, reduced levels of EMC lead to increased failure in membrane integration for a membrane protein TM segment that is already highly prone to misintegration. The same behavior was observed for Cx32$^{L81H}$, showing that this effect of EMC is more general (Supplementary Fig. 4b).

Failures in membrane integration can expose TM helices of Cx32 to the ER lumen, as our data show. We thus wondered whether the ER–lumenal chaperone BiP, a major ER chaperone that detects hydrophobic peptide stretches[35], would be able to recognize misintegrated TM helices of Cx32$^{L90H}$. This would indicate a cooperation of ER–lumenal and membrane integrated chaperone systems on membrane proteins with aberrant topologies. Indeed, BiP bound significantly stronger to Cx32$^{L90H}$, as well as to Cx32$^{L81H}$, than to Cx32$^{wt}$ (Fig. 3d and Supplementary Fig. 4c). To directly query binding of BiP to Cx32 TM sequences, we used a previously established reporter system, where TM segments of interest are fused to the BiP-inert antibody C$_L$ domain[12] (Fig. 3e). In agreement with their predicted free energies for membrane integration (Fig. 1b), we found TM segment 1 of Cx32 to be mostly membrane-integrated, whereas TM segment 2, independent of the additional presence of the L90H mutation, was mostly entering the ER lumen (Fig. 3e and Supplementary Fig. 4d). Of note, these data also reveal that TM segment 2 of Cx32, even in its wild type form, would be unstable in the membrane in isolation and is stabilized in the membrane in the context of Cx32; and furthermore, that the L90H mutation tips the balance, not allowing its complete membrane integration any more. Whereas only very weak interaction of TM segment 1 and BiP was observed, TM segment 2 (wild type and L90H) strongly bound to BiP, corroborating that these TM sequences can directly be recognized by the ER–lumenal chaperone BiP (Fig. 3f).

**The E3 ligase gp78 mediates connexin mutant degradation.** Our data show that mutations in Cx32 can lead to membrane misintegration and rapid degradation via ERAD. We thus wondered which ERAD-associated E3 ligases are involved in degrading Cx32 mutants and assessed the effect of dominant negative mutants of Hrd1 and gp78, two key E3 ligases involved in ERAD[36], on the degradation of Cx32$^{L90H}$. Whereas a slight stabilization of Cx32$^{L90H}$ was detected upon overexpression of the inactive Hrd1$^{C291S}$ mutant[37] (Supplementary Fig. 5), overexpression of the inactive gp78 RING finger mutant C341/378S[38] had a much stronger stabilizing effect on Cx32$^{L90H}$ (Fig. 4a). Furthermore, overexpression of this mutant reduced Cx32$^{L90H}$

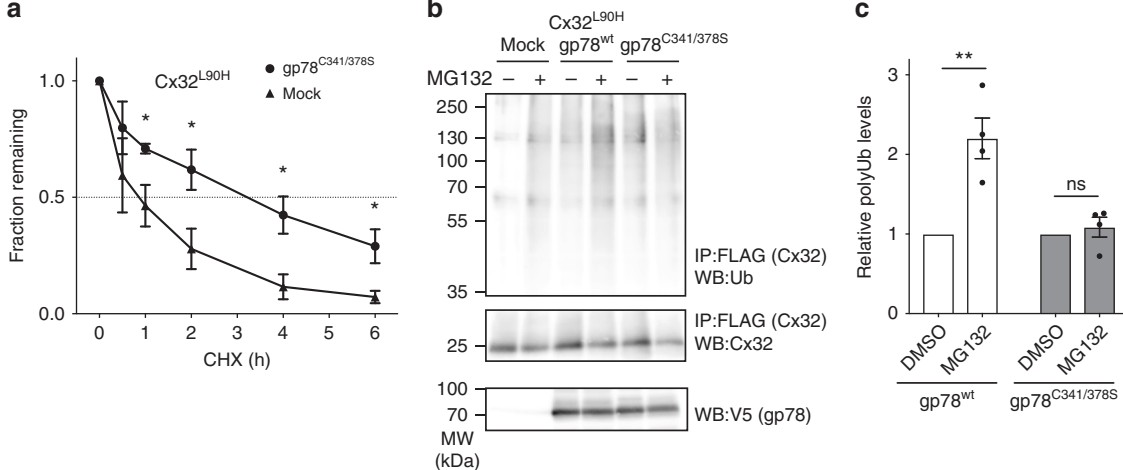

**Fig. 4** gp78 mediates Cx32 degradation. **a** Overexpression of a nonfunctional gp78 mutant (C341/378S) significantly decelerates Cx32$^{L90H}$ degradation (mean ± SEM, $N \geq 3$, *$P$ value < 0.05, two-tailed Student's $t$ tests). **b** Mutant gp78 (C341/378S) overexpression inhibits polyubiquitination of Cx32. FLAG-tagged Cx32$^{L90H}$ was expressed either alone or in the presence of V5-tagged wild type gp78 or gp78$^{C341/378S}$ as indicated. Samples were immunoprecipitated against FLAG (Cx32$^{L90H}$) and blotted against Cx32 or ubiquitin (Ub). Where shown, MG132 was added to inhibit proteasomal degradation of Ub-modified Cx32$^{L90H}$. As can be seen from the blots and quantifications in **c**, MG132 leads to increased ubiquitination of Cx32$^{L90H}$ in the absence or presence of gp78$^{wt}$ overexpression, while gp78$^{C341/378S}$ overexpression inhibits Cx32$^{L90H}$ polyubiquitination. **c** Quantifications of the data shown in **b** (mean ± SEM, $N = 4$, ns: nonsignificant, **$P$ value < 0.01, two-tailed Student's $t$ tests)

ubiquitination upon inhibition of the proteasome with MG132 (Fig. 4b, c). Taken together, this argues that gp78 plays a major role in the ERAD of Cx32[L90H] as a membrane protein with an aberrant topology.

## Discussion

Due to its complexity, membrane protein folding can fail during multiple stages. Soluble domains may misfold, and be taken care of by the general cellular chaperone system as well as by specific factors that recruit this system to the membrane[39,40]. Furthermore, misfolding and aggregation within the membrane can occur, which can have immediate consequences for the etiology of diseases. As an example, a very convincing correlation between protein instability/defects in ligand binding caused by mutations and severity of clinical disease markers was recently shown for PMP22[41]. Lastly, individual TM helices may fail to be integrated into the membrane either due to mutations[18] or due to structural/functional requirements that compete with TM segment hydrophobicity[17,42]. Within this study, we investigated the effects of incorrect topology formation/membrane integration on membrane protein quality control. Although misintegration of TM helices can drastically reduce a protein's half-life[43], molecular mechanisms of how this occurs have remained unclear. Incomplete membrane integration will not only expose TM helices to an inadequate environment, e.g., the ER lumen, but will also leave the remainder of integrated TM segments unpaired. Such a scenario poses a complex challenge to the cellular quality control systems—and indeed we find membrane-embedded factors (Cnx/EMC) as well as the ER–lumenal chaperone BiP to act on Cx32 mutants with compromised membrane integration. Interestingly, Cnx and EMC have been shown to form a complex[33] and may thus act on membrane protein clients in a coordinated manner; also on nonglycosylated ones like Cx32, where recognition of TM helices by Cnx is likely to be involved[44–46]. Since Cx32 forms hexamers, oligomerization needs to be regulated and controlled, and the factors we identify for its biogenesis might also be involved in this process. For the EMC itself, our data provide biochemical evidence that it can aid in integration of less hydrophobic TM segments of multi-pass TM proteins. This is consistent with and extends recent studies where more hydrophilic TM proteins were found to recruit EMC to the translocon[32], a direct role for EMC in integrating low-hydrophobicity TM regions of tail-anchored proteins[31] and a proposed role of EMC as a membrane chaperone[32,34,47]. Indeed, one of the first functions described for EMC was the stabilization of early intermediates in acetylcholine receptor biogenesis in *Caenorhabditis elegans*[34]. The effects we observe are modest, but if indeed EMC predominantly acts co-translationally[32], are to be expected from our steady-state analysis under partial knockdown conditions. Cx32 mutants established during this study now provide a biomedically relevant model system to further dissect EMC functions in future studies. These will have to address how the different functions of EMC are regulated, which role its individual subunits play in these processes and how functions are shaped by interactions with other proteins and within the complex itself. This is of particular relevance, since a very recent study has shown that the EMC can replace the Sec61 translocon in integrating the first TM helix of many G protein coupled receptors as type I TM proteins[48], and thus define their downstream topology. Although the common denominator is membrane protein biogenesis, a multitude of molecular functions have thus been associated with the EMC. The structural complexity of the human EMC, which is composed of ten subunits[31], is compatible with these multiple functions. The large ER–lumenal domains in EMC7, 10, and in particular in EMC 1, but also its cytoplasmic soluble components (EMC2, 8, and 9), suggest functional interactions on both sides of the ER–membrane. Protecting nascent membrane proteins against premature degradation and acting as a quality control hub[32] as well as a possible insertase role for downstream TM segments[48] are very plausible functions that have been discussed for the EMC and are supported by our study.

Our data show that if membrane integration fails, BiP can recognize misintegrated TM regions of a multipass TM protein. This is consistent with previous findings that BiP can bind to the mislocalized ER–lumenal TM region of a single-pass TM protein, the T cell receptor α chain[12], and underlines the close connection of different chaperone systems at the ER. Since BiP generally binds extended hydrophobic stretches in the ER[49], these findings also indicate that the analyzed TM helices, once outside of the membrane, become unstructured. It is likely that cytosolic Hsp70 can act analogously if TM segments became exposed to the cytoplasm. In this regard it is noteworthy that ER-localized J proteins with a cytoplasmic J-domain as the Hsp70-recuitment site have just recently been described to be involved in ion channel assembly[40]. In this case, however, J-proteins were found to act independently of Hsp70. Interestingly, our mass spectrometry data reveal ERdj3 (DNAJB11), one of the ER–lumenal BiP Hsp40 co-chaperones[50], to bind to Cx32 (Fig. 3a), which may suggest a role for this J-protein in Cx32 biogenesis.

Finally, our data show that the ubiquitin E3 ligase gp78 plays a major role in the degradation of a Cx32 mutant with aberrant topology. This finding is particularly interesting as membrane tethered ERAD substrates can lose their dependence on Hrd1 and become gp78-dependent[51]. A misintegrated TM protein might very well resemble such a membrane-tethered substrate. How the Cnx/EMC/BiP systems are coupled to this pro-degradation E3 ligase emerges as an important question and other ER chaperones are likely to be involved[52]. Our study now provides a basis for assessing in more detail how the triage decision between chaperoning and degradation for misintegrated membrane proteins occurs on a molecular level (Fig. 5). Further insights into this process will have an immediate impact on our understanding of how cells produce membrane proteins—and how this process fails in human disease.

## Methods

**DNA constructs**. Cx32 cDNA was obtained from Origene and constructs were cloned into a pSVL vector (Amersham) for mammalian expression. All mutations/insertions were generated via site-directed mutagenesis, using a pair of complementary mutagenic primers and Pfu polymerase (Promega)/DpnI (NEB) or via PCR amplification, restriction and ligation with T4 ligase (NEB, for FLAG tags). For a complete list of primers please see Supplementary Table 1. Glycosylation sites (Asn-Val-Thr), and the FLAG tag were inserted flanked, or preceded, respectively, by a (Gly-Ser)$_2$ linker sequence. C$_L$ domain containing constructs were synthesized by GeneArt (Thermo Fischer) with each individual Cx32 TM segment combined from the sequences deposited in Uniprot and predicted with dgpred.cbr.su.se[17], to avoid artificially shortening TM sequences. TM segment 1 was inverted due to its type II topology. The BiP expression plasmid has been described previously[53]. The gp78-V5/Hrd1-His$_6$ expression plasmids, using a pcDNA3.1 backbone, were cloned and mutated as described above. All constructs were verified by sequencing.

**Cell culture and transient transfections**. HEK293T and COS-7 cells (both from ECACC) were cultured in Dulbecco's Modified Eagle's Medium containing L-Ala-L-Gln (AQmedia, Sigma-Aldrich) supplemented with 10% (v/v) fetal bovine serum (Biochrom) and a 1% (v/v) antibiotic–antimycotic solution (25 μg/ml amphotenicin B, 10 mg/ml streptomycin, and 10,000 units of penicillin; Sigma-Aldrich). Cells were maintained at 37 °C in a humidified 5% CO$_2$ atmosphere. For transient transfections, HEK293T cells were seeded in either poly-D-lysine-coated P35 or P60 (Corning), or P100 dishes (Techno Plastic Products). Twenty-four hours after seeding, cells were transfected with GeneCellin (BioCellChallenge) and 2 μg (p35) or 4 μg (p60) DNA for another 24 h according to manufacturer's instructions. For siRNA-mediated knockdown, 24 h after seeding, cells were transfected with Lipofectamine RNAiMAX (Thermo Scientific) with 25 nM of either control siRNA (Thermo Scientific, 4390843), or a mix of siEMC5 (Thermo Scientific, s41129) and siEMC10 (Thermo Scientific, s49611). Twenty-four hours after siRNA transfection, the plasmid DNA was transfected as described above.

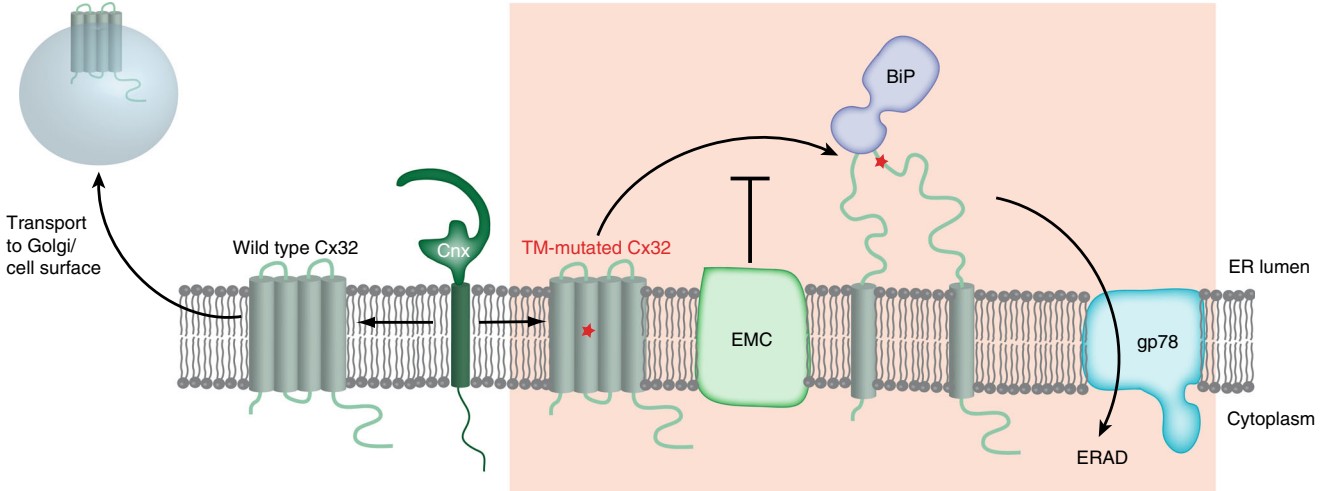

**Fig. 5** A model for the biogenesis and quality control of multipass TM proteins. A red asterisk indicates a mutation that leads to membrane misintegration for Cx32. Chaperones and quality control factors identified in this study to act on Cx32 are shown

For translational arrest, cells were incubated with 50 µg/ml cycloheximide (CHX, Sigma-Aldrich) for different times as shown in the figures. Where indicated, cells were incubated with 10 µM MG132 (Sigma-Aldrich). When both inhibitors were included, an initial 3 h incubation with MG132 was performed, followed by CHX and MG132 co-incubation for the times indicated in the figures. To vehicle-control for CHX and MG132 incubations, DMSO was used at the same volume like the respective inhibitors, always below 0.1% (v/v).

**Cell lysis and deglycosylation assays**. All steps were performed on ice or with ice-cold solutions, unless otherwise stated. HEK293T cells were washed twice with PBS and lysed in NP40 lysis buffer (50 mM Tris/HCl, pH 7.5, 150 mM NaCl, 0.5% (v/v) NP40, 0.5% (w/v) NaDOC) supplemented with 1× protease inhibitor, w/o EDTA (Roche) for 10 min. Samples were centrifuged for 15 min at 15,000$g$ and 4 °C, and the supernatant was used. Supernatants destined for immunoblots were supplemented with Laemmli with 2-mercaptoethanol (β-ME), unless stated otherwise, followed by a 30 min incubation at 37 °C. For nonreducing SDS-PAGE, cells were washed and lysed in the same conditions, except for the addition of 20 mM N-ethylmaleimide (NEM) to the PBS and NP40 buffer. After centrifugation, supernatants were split into two tubes with Laemmli, one supplemented with β-ME, another with NEM.

For deglycosylation assays, samples were digested for 1 h at 37 °C with EndoH or PNGaseF (NEB), according to manufacturer's instructions but w/o boiling for experiments were Cx32 was analyzed. Finally, digestion products were supplemented with Laemmli containing β-ME.

**Antibodies**. Primary antibodies and dilutions used for immunoblots were the following: polyclonal rabbit anti-connexin32 (Sigma-Aldrich, C3595), at 1:500; mouse monoclonal (M2) anti-FLAG (Sigma-Aldrich, F1804), at 1:1000; mouse monoclonal (C8.B6) anti-calnexin (Chemicon, MAB3126), at 1:1000; mouse monoclonal (B-6) anti-Hsc 70 (Santa Cruz, sc-7298) at 1:1000; polyclonal goat anti-mouse lambda (Southern Biotech, 1060-01) at 1:250; mouse monoclonal (BMG-His-1) anti-His$_6$, conjugated to HRP (Sigma-Aldrich, 11965085001) at 1:1000; mouse monoclonal (SV5-Pk1) anti-V5 (Abcam, ab27671) at 1:1000; mouse monoclonal (P4D1) anti-ubiquitin (Santa Cruz, sc-8017) at 1:500; rabbit monoclonal anti-EMC4 (Abcam, ab184544) at 1:10,000; rabbit polyclonal anti-EMC5 (Abcam, ab174366) at 1:250; rabbit monoclonal anti-EMC10 (Abcam, ab181209), at 1:10,000. The BiP antiserum has been described previously[54] and was used at 1:1000. Antibodies were diluted in 5% (w/v) skimmed milk in TBS with 0.05% (v/v) Tween-20. Species-specific secondary antibodies were from Santa Cruz, and used at a 1:10,000 dilution in 5% (w/v) skimmed milk in TBS with 0.05% (v/v) Tween-20.

Antibodies and dilutions used for immunofluorescence were mouse monoclonal (M2) anti-FLAG (Sigma-Aldrich, F1804) at 1:500; rabbit monoclonal (D6B1) anti-GM130 (Cell Signaling Technology, #12480) at 1:3200, used as a Golgi marker; donkey polyclonal anti-mouse IgG, conjugated with TexasRed (Thermo Fischer, PA1-28626) was used at 1:300; anti-rabbit IgG, conjugated with Alexa Fluor® 488 (Cell Signaling Technology, #4412) at 1:500. Rabbit monoclonal (C81H6) anti-PDI, conjugated with Alexa Fluor® 488 (Cell Signaling Technology, #5051), at 1:50, was used as ER marker.

**Immunoprecipitations and immunoblots**. Immunoprecipitation samples were prepared by lysing cells, grown on a P60 dish, in digitonin buffer (50 mM Tris-HCl pH 7.5, 150 mM NaCl, 1% (v/v) digitonin (Sigma-Aldrich), 1× protease inhibitors) followed by a centrifugation step at 15,000$g$ for 15 min at 4 °C. A small aliquot was

supplemented with Laemmli supplemented with β-ME and used for whole cell lysate input controls. The remaining sample was incubated for three hours with 30 µl of mouse monoclonal (M2) anti-FLAG affinity gel (Sigma-Aldrich, A2220) beads under rotation at 4 °C. Beads were washed three times with 50 mM Tris-HCl pH 7.5, 150 mM NaCl, 0.5% (v/v) digitonin. After washing, proteins were eluted from beads with Laemmli buffer (supplemented with 2% (v/v) β-ME) for 30 min at 37 °C.

BiP co-immunoprecipitations were performed as described above, with the exception of lysis in NP40 buffer supplemented with apyrase (Sigma, A6132) and washing steps in NP40 buffer with 400 mM NaCl. For polyubiquitination blots, FLAG immunoprecipitations were performed in the same NP40 buffer (without apyrase addition). $C_L$-TM construct immunoprecipitations were performed in NP40 buffer for 1 h with 2 µg of polyclonal goat anti-mouse lambda antibody (Southern Biotech, 1060-01), and then rotated for another hour with Protein A/G PLUS-Agarose beads (Santa Cruz, sc-2003). Proteins were eluted for 5 min at 95 °C with Laemmli buffer.

Samples were run on 10 to 15% SDS-PAGE gels. Proteins were blotted onto PVDF membranes (Biorad) overnight at 4 °C. After blocking with 5% (w/v) milk in TBS with 0.05% Tween-20 (Biorad), membranes were incubated with primary antibodies. Antibodies were decorated with the respective secondary antibodies coupled to HRP (Santa Cruz) and proteins were detected using Amersham ECL Prime (GE Healthcare) on a Fusion Pulse 6 imager (Vilber Lourmat). Bands were quantified with the Bio-1D software (Vilber Lourmat). Uncropped immunoblots are given in Supplementary Fig. 6.

**Immunofluorescence**. $1.2 \times 10^4$ COS-7 cells were seeded in µ-slides VI$^{0.4}$ (Ibidi) after transfection with 3.6 µg of DNA and Torpedo (Ibidi), and medium was replaced 3 h after seeding, according to the manufacturer's instructions. 48 h after transfection, cells were washed twice with PBS at 37 °C, and fixed and permeabilized with ice-cold methanol (Sigma-Aldrich) at −20 °C for 10 min. After two washing steps with PBS, nonspecific epitopes were blocked with 3% (w/v) BSA (Sigma-Aldrich) diluted in 0.3% (v/v) Triton-X 100 containing PBS. This protocol was used for data in Supplementary Fig. 2a. For all other microscopy experiments, a recently established protocol was used[55]: cells were fixed with glyoxal (Sigma-Aldrich) for 30 min on ice, followed by another 30 min at RT, and then quenched for 20 min with NH$_4$Cl (Sigma-Aldrich). In these, permeabilization and blocking were performed for 15 min with 2.5% BSA in 0.1% Triton-X 100 in PBS. Primary antibody incubations were carried out for 2 h at RT; samples were washed three times with PBS. For fluorophore-conjugated antibodies, samples were handled in the dark during and after incubation. Samples were incubated with secondary antibody for 1 h, followed by one wash with PBS. Nuclei were briefly stained with 0.1 µg/ml DAPI (Sigma-Aldrich), and samples washed three times. Samples were mounted with mounting medium (Ibidi) and imaged on a Leica DMi8 CS Bino inverted widefield fluorescence microscope using either a 63× (NA = 1.40) or a 100× (NA = 1.40) oil immersion objective, and GFP (excitation/bandpass: 470/40 nm; emission/bandpass: 525/50 nm), TXR (excitation/bandpass: 560/40 nm; emission/bandpass: 630/75 nm), or DAPI (excitation/bandpass: 350/50 nm; emission/bandpass: 460/50 nm) dichroic filters. When deconvoluted, z-stacks were recorded with the system-optimized z-size steps, from before the plane where the first punctae structures were seen until after the last plane where they were still detectable. Z-stacked images were then exported to Huygens Essential (SVI) and deconvoluted. Images were analyzed with the LAS X (Leica) analysis software, and analyzed/assembled using ImageJ (NIH), where adjustments were limited to homogenous changes in brightness and contrast over entire images each.

**Quantifications and statistics**. Fractions of glycosylated Cx32 were calculated as the ratio of intensities between glycosylated species and the sum of both glycosylated and nonglycosylated species. In siRNA experiments, glycosylated fractions were each normalized to the siCTRL samples. CHX chase quantifications were performed by normalizing each band intensity to the intensity at 0 h. Half-life calculations were performed by logarithmically linearizing the exponential decay curves, and intercepting this to ln(0.5). Co-immunoprecipitation samples were quantified by normalizing the band intensity of the co-immunoprecipitated protein to the intensity of immunoprecipitated protein. If Cx32 was immunoprecipitated, the sum of monomer and dimer bands was used if not stated otherwise. Polyubiquitination quantifications were performed by subtracting from the signal intensity of FLAG IP samples in each treatment/co-transfection the corresponding signal in samples transfected with a mock plasmid. Total intensities were used, beginning at the molecular weight of FLAG-tagged, monoubiquitinated Cx32 (ca. 35 kDa) until the upper end of the separating gel.

Punctae were counted from deconvoluted TXR filter images using ImageJ (NIH). Briefly, threshold values were set using Otsu's method, and punctae were filtered to be ten pixels or larger in size, corresponding to the average size of plaque-like structures from Cx32wt-FLAG cells.

Analyses were performed using GraphPad (Prism). Where $P$ values are given, data were analyzed with two-tailed Student's $t$ tests.

**Mass spectrometry data acquisition**. HEK293T cells were seeded in P100 plates and transfected with 10 µg of FLAG-tagged Cx32. Immunoprecipitations were performed in digitonin buffer as described above, but using a rabbit monoclonal anti-FLAG (Sigma-Aldrich, F7425) and a rabbit IgG isotype control (Thermo Scientific, 10500C), with a total of three replicates each. Two additional washing steps were performed without digitonin. Proteins were digested, eluted, desalted, and purified as previously described[27]. Nanoflow liquid chromatography-mass spectrometry (MS)/MS analyses were performed with an UltiMate 3000 Nano HPLC system (Thermo Scientific) coupled to an Orbitrap Fusion mass spectrometer (Thermo Scientific). Peptides were loaded on a trap column (Acclaim C18 PepMap100 75 µm ID × 2 cm) with 0.1% TFA, then transferred to an analytical column (Acclaim C18 PepMap RSLC, 75 µm ID × 50 cm, 0.1% FA) heated at 50 °C and separated using a 105 min gradient from 5 to 22% followed by a 10 min gradient from 22 to 32% acetonitrile in 0.1% FA at a flow rate of 300 nl/min. Peptides were ionized using an EASY-ETD/IC source. Orbitrap Fusion was operated in a top speed data dependent mode with a cycle time of 3 s. Full scan (MS$^1$) acquisition (scan range of 300–1500 m/z) was performed in the orbitrap at a resolution of 120,000 and with an automatic gain control (AGC) ion target value of 2e5. Dynamic exclusion of 60 s as well as EASY-IC internal calibration was enabled. Most intense precursors with charge states of 2–7 and a minimum intensity of 5e3 were selected for fragmentation. Isolation was performed in the quadrupole using a window of 1.6 m/z. Fragments were generated using higher-energy collisional dissociation (HCD, collision energy: 30%). The MS$^2$ AGC target was set to 1e4 and a maximum injection time for the ion trap of 50 ms was used (with inject ions for all available parallelizable time enabled). Fragments were scanned with the rapid scan rate.

**MS bioinformatics**. MS raw files were analyzed with MaxQuant software (version 1.5.3.8)[56] with most default settings and a protein database containing human sequences (downloaded May 2017 from Uniprot, taxonomy ID: 9606). Note: Although the target protein Cx32 was FLAG-tagged and occasionally mutated, the original protein sequence was not modified. The following parameter settings were used: PSM and protein FDR 1%; enzyme specificity trypsin/P; minimal peptide length: 7; variable modifications: methionine oxidation, N-terminal acetylation; fixed modification carbamidomethylation. For label-free protein quantification, the MaxLFQ algorithm[57] was used as part of the MaxQuant environment: (LFQ) minimum ratio count: 1; peptides for quantification: unique and razor. The match between runs option was enabled (match time window: 0.7 min, alignment time window: 20 min). Statistical analysis was performed in Perseus (version 1.5.3.2)[58]. Proteins identified only by site, reverse hits or potential contaminants were removed. LFQ intensities were log2 transformed and the matrix was separated to submatrices for each comparison (wt vs. wt iso, L90H vs. L90H iso, I203N vs. I203N iso, wt vs. L90H and wt vs. I203N). Data were then filtered for at least two valid values in at least one replicate group. Then, missing values were imputed from normal distribution (width: 0.3, down shift: 1.8 standard deviations, mode: separately for each column). The replicate groups were compared via a two-sided, two-sample Student's $t$ test (S0 = 0, permutation-based FDR method with FDR = 0.05 and 250 randomizations). Enrichment values and corresponding −log10 $P$ values were plotted. The mass spectrometry proteomics data have been deposited to the ProteomeXchange Consortium (http://proteomecentral.proteomexchange.org) via the PRIDE partner repository[59].

**Sequence analyses and structural modeling**. Structural modeling for human Cx32 was performed in iTasser[60] taking the chain F of Cx26 as a template (PDB ID: 2zw3). Template and model sequences/structures share 49% identity, a coverage of 71%, RMSD of 0.72 Å, and a TM-score of 0.703. The normalized Zscore for the model was 2.68. After model generation, individual Cx32 monomers were superimposed on the Cx26 hexamer. The achieved Cx32 hexamer model was energy minimized using Yasara Structure (www.yasara.org), and images generated using Chimera (UCSF). TM regions were annotated by using dgpred.cbr.su.se[17]. The same tool was used to predict the $\Delta G_{app}$ for helix insertions using the full protein scan mode.

**Reporting Summary**. Further information on experimental design is available in the Nature Research Reporting Summary linked to this Article.

## Data availability
Data supporting the findings of this manuscript are available from the corresponding author upon reasonable request. A Reporting Summary for this Article is available as a Supplementary Information file. Mass spectrometry proteomics data from this study have been deposited to the ProteomeXchange Consortium (http://proteomecentral.proteomexchange.org) with the dataset identifier PXD009189.

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

## Acknowledgments

We thank Katja Bäuml for excellent assistance with mass spectrometry and Eva M. Trautmann for experimental assistance during the early stages of the project. The BiP expression plasmid and antiserum were a kind gift from Linda Hendershot, St. Jude Children's Research Hospital. M.S. was supported by the German Academic Scholarship Foundation and the Wellcome Trust. Z.R.Z. gratefully acknowledges funding by the National Natural Science Foundation of China (Grant no. 31670780). S.A.S. greatfully acknowledges funding by the European Research Council (ERC), the European Union's Horizon 2020 research and innovation program (grant agreement no. 725085, CHEM-MINE, ERC consolidator grant). M.J.F. is a Rudolf Mößbauer Tenure Track Professor and as such gratefully acknowledges funding through the Marie Curie COFUND program and the Technical University of Munich Institute for Advanced Study, funded by the German Excellence Initiative and the European Union Seventh Framework Program under Grant Agreement 291763. Funding of our work on Cx32 by the Fritz Thyssen foundation and on membrane proteins by a German Research Foundation (DFG) research grant, number FE1581/1-1, is gratefully acknowledged.

## Author contributions

M.J.F. conceived the study, mass spectrometry experiments were performed by M.S.; J.C., N.B., K.M.B. and C.P.A. performed all other experiments. The data were analyzed by J.C., M.S., N.B., K.M.B., C.P.A., Z.R.Z., S.A.S. and M.J.F. The paper was written by J.C. and M.J.F. with input from M.S., N.B., K.M.B., C.P.A., Z.R.Z., and S.A.S.

## Additional information

**Competing interests:** The authors declare no competing interests.

