## [Peer Review File · Nature Communications]

Reviewers' comments:

Reviewer #1 (Remarks to the Author):

The first major claim of the paper is that the polytopic, hexamer-forming membrane protein connexin 32 (Cx32) has one transmembrane segment (TMS) of lower hydrophobicity that is particularly sensitive to the effects of mutations resulting in the presence of polar or charged side chains in the TMS. In fact, the authors demonstrate that such changes can tip the balance and interfere with membrane integration. The first two main and supplementary Figures make this point rigorously and by presenting data of the highest quality. Second, the authors claim to have mapped the chaperone network handling such membrane proteins with challenging TMSs and leading to their degradation in the event of integration problems such as provoked by the investigated disease-causing mutation in TMS2 of Cx32. The key complex identified here is the EMC complex, which is currently receiving a lot of attention as the field begins to characterise its function. Given the many subunits that make up the EMC and their high degree of evolutionary conservation the functional scope of this complex is bound to be complex and not likely to be confined to one exclusive activity. This is where the manuscript makes a substantial contribution by clearly demonstrating a role in the biogenesis of Cx32. Indeed, the work presented here provides elegant experiments that mechanistically extend a systematic study in yeast identifying multispinning membrane proteins as biogenetic clients of the EMC (Shurtleff et al 2018). Hence, I am convinced that this paper will be very influential on the thinking in the field where different hypotheses for the functionality of the EMC have been raised and need to be tested and refined. The last Figure identifies the E3 ligase gp78 as the major enzyme involved in the ubiquitylation of Cx32 exposing TMSs to the ER lumen. Since gp78 has been shown to be involved in the stability of polytopic membrane proteins such as HMG-CoA reductase the manuscript adds an important class of substrates and an entry point for more mechanistic work. Taken together, the manuscript presents a strong data set nicely supporting important conclusions about the mechanistic consequences of disease-causing mutations. It provides the field with a fresh perspective and avenues to follow for further understanding of a central yet still somewhat mysterious machinery in the ER.

Minor points:

Fig. 2a/p. 7: I find the statement that the mutant Cx32L90H partially co-localizes with PDI not well supported by the data. The foci formed by the altered protein are not close to the plasma membrane, that is true. Yet, co-localization would mean that the foci are PDI-positive, which I do not see from the provided images. Other ER markers should be tried.

Fig. 3d: There seem to be more SDS-resistant dimers for Cx32 L90H. Hence I am not sure whether the increase in EMC component binding is reflected by a stronger avidity of this species to the EMC or its contribution to the total Cx32, which may then be higher than for the wt (how this upper band was handled in the quantification wasn't clear to me from the quantification description in the Methods section).

Reviewer #2 (Remarks to the Author):

Feige and co-workers report on the membrane integration of a medical interesting target. Connexin 32 (Cx32) is a membrane protein related to Charcot-Marie-Tooth disease. The authors show that a single point mutation cause partial misintegration leading to transport defects and accelerated degradation. Then, they applied proteomic approaches to search for chaperone partners and found calnexin and EMC 10 as closely related to both wild type and mutant Cx32. They also investigated the role of ER luminal chaperone BiP, and finally, they demonstrated that gp78 but not Hrd1 mediates Cx32 degradation, providing a likely route for protein degradation. The findings are interesting but the relevance and interpretation do not appear fully clear to me.

General comments, followed by specific comments/criticisms are below.

General:

1- The mutation selected to be studied along the manuscript (L90H) produces only partial misintegration (Fig. 1), which probably hampers to obtain more clear results in the study. Why the authors did not search for mutations with stronger impact on membrane integration? In fact, they found that double mutant L83R/L90H is fully glycosylated at loop2 (Fig. 1f), why not to use this double-mutant in the rest of the work? The inclusion of this mutant (or any other with similar integration behavior) would likely provide more clear data.

Since L90H mutation affects integration of TM2 and TM3, a mutation in TM3 generating loop2 glycosylation is highly recommended.

2- Fig. 3: Vulcano plot for Cx32(L90H)-FLAG mutant should be shown. In fact, EMC 4 is not highlighted in Fig. 3a, whilst co-immunoprecipitates both with Cx32wt and Cx32(L90H) mutant. In the same direction, both EMC components tested in the immunoprecipitation experiments (4 and 10) bind more efficiently to the mutant protein, this fact should be observed in the mass-spectrometry experiments (see also next point).

3- The authors should provide the proteomic data, both the list of identified proteins as well as the quantitative values for all of them as supplementary information.

4- Regarding the ERAD-associated E3 ligases, Hrd1 data should be plotted.

Specific:

1- The authors should show Cx32 sequence and place the glycosylation acceptor sequons and the FLAG tag in the expressed chimera. This assists the reader in comparatively evaluating the glycosylation efficiencies found.

2- Cx32(L90H) mutant dimerizes more efficiently than wild-type protein (Figs. 1 and 2). Would this be the reason for the observed effects? The authors should comment on that in the body text.

3- Explain the differences between EndoH and PNGaseF performance in the body text.

4- Fig. 3a: explain what is 'isotype control'.

5- Fig. 3e: TM1 inverted sequence was used as insertion control. I assume that the sequence was inverted because the orientation of this first TM segment is opposite in the full-length protein. Can the authors prove that an inverted sequence implies a reversed topology in the ER membrane?

6- Fig. 4b: It is difficult to me to appreciate differences in the WB:Ub image. In addition, the apparently lower level of Cx32 in lane '6' (middle panel) could be responsible for the lower level of intensity in the ubiquitinated bands in this lane '6' compared to lane '4' (upper panel).

7- In line with Nat. Commun. policy 3b, 3c, 3d and 4c bar graphs should show individual data points.

Reviewer #3 (Remarks to the Author):

This study addresses a very interesting and ill-understood problem in contemporary cell biology: How do membrane proteins integrate into a membrane, how do disease-causing mutations affect this process and how does a cell cope with aberrant integration? Focusing on connexin 32, a well-known model protein where multiple hereditary disease-causing mutations are known, the authors provide a very comprehensive analysis detailing different steps of this process. First, they use glycosylation mapping to show that three mutations to polar residues within TM2, but not certain mutations in other TMDs, alter topogenesis of connexin 32. They show that altered topogenesis is associated with partial ER-retention as well as faster degradation and plaque formation of the mutants. Second, they find that the L90H mutant binds more strongly to two chaperones known to be involved in membrane protein biogenesis (EMC, calnexin). Indeed, the partial knock-down of EMC subunits increases aberrant loop 2 glycosylation of connexin 32 which is taken as evidence for a corrective function of the EMC complex. Further, there is stronger EMC and BIP binding to the L90H mutant. Third, they provide evidence for involvement of the E3 ligase gp78 in L90H degradation.

This is a very elegant study addressing key steps in the mechanism of membrane protein biogenesis. The results have very strong explanatory power when it comes to the mechanism of how disease-causing point mutations affect membrane protein integration. By linking these different steps in one study, I believe this is a major advance in our understanding of membrane protein biogenesis and its alteration by sequence aberrations.

I have only a few comments:

1. The increases in the glycosylated fraction in Fig. 3C and Fig. S3B is only 8 % and 12 %, respectively. Albeit statistically significant, these increases are really small and one wonders whether they are biologically significant. The authors might consider a somewhat more cautious wording here.
2. Fig. 4B and C is difficult to understand. I suggest a better explanation in the legend.
3. Can they explain the somewhat faster migration of the N-term constructs relative to the other constructs in Fig. 1D?

Dear reviewers,

We would like to thank you very much for the positive evaluation as well as the constructive criticism on our study. Please find below a point-by-point reply where we have addressed your concerns by additional experiments as well as by editing the text and figures of our manuscript. We hope that our reply addresses all your concerns adequately and would like to thank you again for the time you have invested in reviewing our manuscript.

Reviewer 1:

The first major claim of the paper is that the polytopic, hexamer-forming membrane protein connexin 32 (Cx32) has one transmembrane segment (TMS) of lower hydrophobicity that is particularly sensitive to the effects of mutations resulting in the presence of polar or charged side chains in the TMS. In fact, the authors demonstrate that such changes can tip the balance and interfere with membrane integration. The first two main and supplementary Figures make this point rigorously and by presenting data of the highest quality. Second, the authors claim to have mapped the chaperone network handling such membrane proteins with challenging TMSs and leading to their degradation in the event of integration problems such as provoked by the investigated disease-causing mutation in TMS2 of Cx32. The key complex identified here is the EMC complex, which is currently receiving a lot of attention as the field begins to characterise its function. Given the many subunits that make up the EMC and their high degree of evolutionary conservation the functional scope of this complex is bound to be complex and not likely to be confined to one exclusive activity. This is where the manuscript makes a substantial contribution by clearly demonstrating a role in the biogenesis of Cx32. Indeed, the work presented here provides elegant experiments that mechanistically extend a systematic study in yeast identifying multispinning membrane proteins as biogenetic clients of the EMC (Shurtleff et al 2018). Hence, I am convinced that this paper will be very influential on the thinking in the field where different hypotheses for the functionality of the EMC have been raised and need to be tested and refined. The last Figure identifies the E3 ligase gp78 as the major enzyme involved in the ubiquitylation of Cx32 exposing TMSs to the ER lumen. Since gp78 has been shown to be involved in the stability of polytopic membrane proteins such as HMG-CoA reductase the manuscript adds an important class of substrates and an

entry point for more mechanistic work. Taken together, the manuscript presents a strong data set nicely supporting important conclusions about the mechanistic consequences of disease-causing mutations. It provides the field with a fresh perspective and avenues to follow for further understanding of a central yet still somewhat mysterious machinery in the ER.

Minor points:

Fig. 2a/p. 7: I find the statement that the mutant Cx32L90H partially co-localizes with PDI not well supported by the data. The foci formed by the altered protein are not close to the plasma membrane, that is true. Yet, co-localization would mean that the foci are PDI-positive, which I do not see from the provided images. Other ER markers should be tried.

As suggested by the reviewer we have repeated the microscopy experiments – and significantly extended them. First, we changed to a recently established fixation protocol that preserves cellular structures particularly well (Richter *et al.*, *Glyoxal as an alternative fixative to formaldehyde in immunostaining and super-resolution microscopy*, EMBO J, 2018). Additionally, we now have included a Golgi marker (GM130) in our experiments, since we had observed complex glycosylation even for the L90H mutant of Connexin 32. And lastly, we extended our microscopy studies by two additional mutants investigated in our paper: L81H and L83R/L90H. Also using the new fixation protocol, we find Cx32 wild type to form punctae and gap junction plaques as reported in our first submission. L90H indeed remains more complex to interpret: we observe slightly more ER-localization than for the wild type; but also, with the new fixation protocol, the observed dots do not co-localize well with PDI (also not with Cnx, which we tried additionally as another ER marker), and neither with the Golgi marker GM130. Taking this and the Golgi-modification of sugars of Cx32 L90H into account we favor an interpretation of this data that includes i) L90H can traverse through the Golgi, likely reach the cell surface ii) L90H is only slightly more retained in the ER than wild type, possibly rapidly degraded if retained iii) L90H fails to form gap junction plaques which, as opposed to wild type, we do not observe in the microscopy experiments and in detergent solubility experiments. Since L90H expression levels are lower, in agreement with its

more rapid ERAD, a fraction of L90H can likely reach the cell surface (when TM helices are inserted) but not form gap junctions, whereas another fraction is degraded. Our additional experiments show that L81H has a similar behavior like L90H, but its ER retention is more pronounced. The double mutant L83R/L90H, where TM sequences 2 and 3 completely fail to integrate into the membrane, shows complete ER localization. We have included these new data in the main Figure 2 as well as Supplementary Figure 2 and edited the text accordingly.

Fig. 3d: There seem to be more SDS-resistant dimers for Cx32 L90H. Hence I am not sure whether the increase in EMC component binding is reflected by a stronger avidity of this species to the EMC or its contribution to the total Cx32, which may then be higher than for the wt (how this upper band was handled in the quantification wasn't clear to me from the quantification description in the Methods section).

This is indeed an important point. We thus would like to point out that the L81H mutant, which also shows stronger EMC4 binding (Supplementary Fig. 4a of the revision), does show hardly any SDS-resistant dimers (Supplementary Fig. 4a and 4c of the revision). As such, we are hesitant to conclude anything about monomer/dimer binding preferences of EMC. We now mention these findings in more detail in the results. We also have edited the Methods section to provide a better explanation of our quantifications, which always took into account the sum of monomer and dimer species.

Reviewer 2:

Feige and co-workers report on the membrane integration of a medical interesting target. Connexin 32 (Cx32) is a membrane protein related to Charcot-Marie-Tooth disease. The authors show that a single point mutation cause partial misintegration leading to transport defects and accelerated degradation. Then, they applied proteomic approaches to search for chaperone partners and found calnexin and EMC10 as closely related to both wild type and mutant Cx32. They also investigated the role of ER luminal chaperone BiP, and finally, they demonstrated that gp78 but not Hrd1 mediates Cx32 degradation, providing a likely route for protein degradation. The findings are interesting but the relevance and interpretation do not appear fully clear to me. General comments, followed by specific comments/criticisms are below.

General:

1- The mutation selected to be studied along the manuscript (L90H) produces only partial misintegration (Fig. 1), which probably hampers to obtain more clear results in the study. Why the authors did not search for mutations with stronger impact on membrane integration? In fact, they found that double mutant L83R/L90H is fully glycosylated at loop2 (Fig. 1f), why not to use this double-mutant in the rest of the work? The inclusion of this mutant (or any other with similar integration behavior) would likely provide more clear data.

We agree with the reviewer that in some scenarios (e.g. BiP binding to misintegrated TM helices), the completely misintegrated L83R/L90H double mutant could have given clearer results. In others, however, we think partial misintegration allows for a better assessment of quality control factors. The effects of EMC on membrane integration, e.g., would have likely not been observable with such a strong destabilization in the membrane as observed for L83R/L90H. Furthermore, whereas L90H is a naturally occurring disease mutant, L83R/L90H was artificially created by us as a tool. We thus hope that the reviewer understands we mostly focused on L90H (and L81H in the SI material). That being said, we completely agree the L83R/L90H double mutant will be a nice tool for future studies. And in fact, in the revision of our paper we have already used this double mutant more extensively for microscopy experiments (revised Supplementary Figure 2) to show that a mutant that has a

complete failure of membrane integration for TM helix 2 does not leave the ER any more.

Since L90H mutation affects integration of TM2 and TM3, a mutation in TM3 generating loop2 glycosylation is highly recommended.

This is indeed an excellent suggestion, which we happily took up. We have, in addition to the A147D mutant in TM3 that was reported on our first submission now additionally created the mutants V140E and L143P in TM3. However, unfortunately for none of these mutants we could observe glycosylation of loop2. A147D has a predicted free energy of membrane integration of +0.7 kcal/mol, for V140E this value is +0.3 kcal/mol and for L143P it is +0.9 kcal/mol. These values are all well below the values for single point mutations in TM segment 2, which are ca. +1.8 to +2.8 kcal/mol. Thus, it is maybe not surprising that we do not observe misintegration caused by TM3-mutations if we stick to naturally occurring disease-causing mutations. We hope that the reviewer concurs with our interpretation.

2- Fig. 3: Volcano plot for Cx32(L90H)-FLAG mutant should be shown. In fact, EMC4 is not highlighted in Fig. 3a, whilst co-immunoprecipitates both with Cx32wt and Cx32(L90H) mutant. In the same direction, both EMC components tested in the immunoprecipitation experiments (4 and 10) bind more efficiently to the mutant protein, this fact should be observed in the mass-spectrometry experiments (see also next point).

As suggested by the reviewer, we have now included the Cx32(L90H)-FLAG volcano plot in the new Supplementary Fig. 3a. In the proteomic data for L90H, we did not observe EMC4 binding, however. In fact, although the EMC forms a 10-subunit complex, only EMC1, EMC2, EMC7, and EMC10 were detected in mass-spectrometry experiments and among those only EMC10 was found to be enriched in the wild-type samples. This likely can be attributed to common complications in detecting membrane proteins in mass spectrometry experiments. Furthermore, this finding has to be interpreted carefully as the isotype quantification values had been completely imputed from normal distribution. This is a reason why we validated the proteomics-generated

findings by classical immunoblotting combined with statistical analyses to allow for direct comparisons of binding efficiencies between proteins.

3- The authors should provide the proteomic data, both the list of identified proteins as well as the quantitative values for all of them as supplementary information.

We now have included a list as well as quantitative values for chaperones we investigated further (Cnx and EMC10) and that interacted with Cx32 and with Cx32 L90H in the new Supplementary Fig. 3b. We furthermore uploaded the complete and very comprehensive list of proteins identified by mass spectrometry to the PRIDE partner repository with the dataset identifier PXD009189 (username: reviewer35967@ebi.ac.uk / password: hnvxWyXb), which is mentioned in the manuscript mass spectrometry Methods section.

4- Regarding the ERAD-associated E3 ligases, Hrd1 data should be plotted.

As suggested, we now have included new and more complete data on the effect of a dominant negative Hrd1 mutant in the Supplementary material, Fig. S5, which shows a weak effect on Cx32 degradation as previously mentioned in the manuscript. These findings are now also mentioned in more detail in the last section of the results.

Specific:

1- The authors should show Cx32 sequence and place the glycosylation acceptor sequons and the FLAG tag in the expressed chimera. This assists the reader in comparatively evaluating the glycosylation efficiencies found.

We have now included a schematic in Supplementary Figure 1a to make our design more easily accessible to the reader.

2- Cx32(L90H) mutant dimerizes more efficiently than wild-type protein (Figs. 1 and 2). Would this be the reason for the observed effects? The authors should comment on that in the body text.

This is a very good observation, which also reviewer 1 had noticed. We thus would like to draw the attention to Supplementary Figure 4 of the revision, where we assessed Cnx/BiP/EMC binding of Cx32 L81H as well as the effect of EMC

on its membrane integration. We see almost the same binding/effects as for L90H – even though L81H dimerization is significantly reduced in comparison to Cx32 wild type and Cx32 L90H. We thus conclude that the effects we observe are not dominated by dimerization. We now have mentioned these findings in more detail in the manuscript.

3- Explain the differences between EndoH and PNGaseF performance in the body text.

We have now explained the difference between these two enzymes in the main text where EndoH/PNGaseF are mentioned for the first time and excuse this previous omission.

4- Fig. 3a: explain what is 'isotype control'.

We have now included an explanation which isotype control was used in the legend to Figure 3a.

5- Fig. 3e: TM1 inverted sequence was used as insertion control. I assume that the sequence was inverted because the orientation of this first TM segment is opposite in the full-length protein. Can the authors prove that an inverted sequence implies a reversed topology in the ER membrane?

This assumption is entirely correct. To assess the topology of the inverted sequence more thoroughly, we now additionally include a construct with a glycosylation site N-terminal of the TM segment, which should always be ER-luminal if the inverted TM sequence was inserted correctly into the membrane. Indeed, our new data show this site to become glycosylated quantitatively, corroborating the type I topology of the inverted first TM sequence of Cx32 (data are shown in the new Supplementary Fig. 4d).

6- Fig. 4b: It is difficult to me to appreciate differences in the WB:Ub image. In addition, the apparently lower level of Cx32 in lane '6' (middle panel) could be responsible for the lower level of intensity in the ubiquitinated bands in this lane '6' compared to lane '4' (upper panel).

This is indeed a valid point, we therefore performed a quantitative evaluation of the relative ubiquitination. The lower levels for Cx32 we see under all conditions whenever we add MG132, but only for the dominant negative gp78 mutant was this concomitant with reduced ubiquitination.

7- In line with Nat. Commun. policy 3b, 3c, 3d and 4c bar graphs should show individual data points.

In agreement with this suggestion and Nat. Commun. policies we have adjusted all figures accordingly.

Reviewer 3:

This study addresses a very interesting and ill-understood problem in contemporary cell biology: How do membrane proteins integrate into a membrane, how do disease-causing mutations affect this process and how does a cell cope with aberrant integration? Focusing on connexin 32, a well-known model protein where multiple hereditary disease-causing mutations are known, the authors provide a very comprehensive analysis detailing different steps of this process. First, they use glycosylation mapping to show that three mutations to polar residues within TM2, but not certain mutations in other TMDs, alter topogenesis of connexin 32. They show that altered topogenesis is associated with partial ER-retention as well as faster degradation and plaque formation of the mutants. Second, they find that the L90H mutant binds more strongly to two chaperones known to be involved in membrane protein biogenesis (EMC, calnexin). Indeed, the partial knock-down of EMC subunits increases aberrant loop 2 glycosylation of connexin 32 which is taken as evidence for a corrective function of the EMC complex. Further, there is stronger EMC and BIP binding to the L90H mutant. Third, they provide evidence for involvement of the E3 ligase gp78 in L90H degradation.

This is a very elegant study addressing key steps in the mechanism of membrane protein biogenesis. The results have very strong explanatory power when it comes to the mechanism of how disease-causing point mutations affect membrane protein integration. By linking these different steps in one study, I believe this is a major advance in our understanding of membrane protein biogenesis and its alteration by sequence aberrations.

I have only a few comments:

1. The increases in the glycosylated fraction in Fig. 3C and Fig. S3B is only 8 % and 12 %, respectively. Albeit statistically significant, these increases are really small and one wonders whether they are biologically significant. The authors might consider a somewhat more cautious wording here.

In agreement with this suggestion, we have now throughout the entire manuscript used a more cautious wording, and in fact, the small difference was also one of the reasons why we studied the L90H and additionally the L81H

mutant (with similar effects). We also have adjusted how we depict normalization of the glycosylation data in all revised figures to make the change more easily accessible to the reader.

2. Fig. 4B and C is difficult to understand. I suggest a better explanation in the legend.

We excuse for the brevity of the legend and now have significantly extended it to make this Figure more accessible to the reader.

3. Can they explain the somewhat faster migration of the N-term constructs relative to the other constructs in Fig. 1D?

This is indeed a very good observation. The N-term construct runs slightly faster since, as opposed to all other constructs, we did not include an artificial glycosylation site (GSGSNVTGSGS) into this construct, as Cx32 endogenously possesses an N-terminal NWT site. We now have clarified this in the manuscript and also included a schematic of the construct designs in the new Supplementary Figure 1a.

Reviewers' Comments:

Reviewer #1:

Remarks to the Author:

I looked at the revised manuscript and the rebuttal. I think it is now all good.

Reviewer #2:

Remarks to the Author:

The authors have done due diligence and responded satisfactorily my previous comments.